# MobilePrune: Neural Network Compression via *ℓ*_0_ Sparse Group Lasso on the Mobile System

**DOI:** 10.3390/s22114081

**Published:** 2022-05-27

**Authors:** Yubo Shao, Kaikai Zhao, Zhiwen Cao, Zhehao Peng, Xingang Peng, Pan Li, Yijie Wang, Jianzhu Ma

**Affiliations:** 1Department of Computer Science, Purdue University, West Lafayette, IN 47907, USA; shao111@purdue.edu (Y.S.); peng272@purdue.edu (Z.P.); panli@purdue.edu (P.L.); 2Department of Computer Science, Indiana University at Bloomington, Bloomington, IN 47405, USA; kkai_zhao@yeah.net; 3Department of Computer Graphics, Purdue University, West Lafayette, IN 47907, USA; cao270@purdue.edu; 4Institute for Interdisciplinary Information Sciences, Tsinghua University, Beijing 100190, China; xingang.peng@gmail.com; 5Institute for Artificial Intelligence, Peking University, Beijing 100871, China

**Keywords:** mobile computing, model compression, pruning network, deep learning, convolutional neural network

## Abstract

It is hard to directly deploy deep learning models on today’s smartphones due to the substantial computational costs introduced by millions of parameters. To compress the model, we develop an $\ell_{0}$-based sparse group lasso model called MobilePrune which can generate extremely compact neural network models for both desktop and mobile platforms. We adopt group lasso penalty to enforce sparsity at the group level to benefit General Matrix Multiply (GEMM) and develop the very first algorithm that can optimize the $\ell_{0}$ norm in an exact manner and achieve the global convergence guarantee in the deep learning context. MobilePrune also allows complicated group structures to be applied on the group penalty (i.e., trees and overlapping groups) to suit DNN models with more complex architectures. Empirically, we observe the substantial reduction of compression ratio and computational costs for various popular deep learning models on multiple benchmark datasets compared to the state-of-the-art methods. More importantly, the compression models are deployed on the android system to confirm that our approach is able to achieve less response delay and battery consumption on mobile phones.

## 1. Introduction

Deep neural networks (DNNs) have achieved tremendous success in many real-world applications. However, the computational cost of DNN models significantly restricts their deployment on platforms with limited computational resources, such as mobile devices. To address this challenge, numerous model compression algorithms have been proposed to reduce the sizes of DNN models. The most popular solution is to prune the weights with small magnitudes by adding $\ell_{0}$ or $\ell_{1}$ penalties [1,2,3,4]. The non-zero weights selected by these methods are randomly distributed and do not reduce the memory consumption due to the matrix operations widely adopted in nowadays deep learning architectures as shown in Figure 1(b.2). The implementation of such a non-structured sparse matrix in cuDNN [5], which is the Basic Linear Algebra Subroutines library used by deep learning models, has similar memory consumption as the original matrix without pruning, as shown in Figure 1(b.2),(c.2). To overcome the problem, structured pruning models [6,7,8,9] are proposed to enforce group sparsity by pruning a group of pre-defined variables together. By tying the weights connecting to the same neuron together, these approaches are able to prune a number of hidden neurons to reduce the sizes of weight matrices and benefit General Matrix Multiply (GEMM) used in cuDNN as shown in Figure 1(b.3),(c.3). However, one of the main problems of the structured methods is that they do not consider and take advantage of the hardware accelerator architectures.

In this paper, we observe three key factors that could lead to an extremely compact deep network. First, we observe that most modern deep learning architectures rely on the General Matrix Multiply (GEMM) functions implemented in the cuDNN package. We, therefore, propose a new network compression algorithm to harmonize the group selections in the structured penalty and the implementation of GEMM in cuDNN as well as the hardware accelerator using sparse systolic tensor array [10,11]. Figure 1 demonstrates the basic rationale of our observation. In comparison to the pruned model in Figure 1(c.3),(c.4) needs additional sparsity within the remaining groups, which could be utilized by the sparse systolic tensor array for hardware acceleration.

The second observation is that recent studies [12,13] have demonstrated that $\ell_{0}$ norm is the best sparsity-inducing penalty compared to lasso $\ell_{1}$ [14], elastic net [15], SCAD [16], and MCP [17] models. Remarkably, even current $\ell_{0}$ optimization techniques can not achieve the global optimal solution, these $\ell_{0}$-based methods with sub-optimal solutions still significantly outperform other sparsity norms, which can be solved to global optima [13,18]. Hence, the community has shown great interest in using $\ell_{0}$ norm to compress the large-scale DNN models [8,19,20,21,22,23,24]. However, some $\ell_{0}$-based DNN pruning approaches [8,25,26] rely on different relaxation strategies to conquer the non-convex and non-differentiable challenges, which does not fully exploit the strength of $\ell_{0}$ regularization. The other methods [19,20,21,22,23] rely on the alternating direction method of multipliers (ADMM) whose global convergence has not been proved so far when applied to $\ell_{0}$ norm optimization [27,28,29].

Third, most of these algorithms are designed originally for mobile platforms, as the computational resources are relatively rich for desktop applications. However, few of them have been deployed on real mobile systems to test the running time and energy consumption to verify their assumptions.

Therefore, we develop the very first algorithm in this paper, named MobilePrune, which is able to solve the optimization of $\ell_{0}$ sparse group lasso regularization in an exact manner. The main technical contribution is that we solve the proximal operators for $\ell_{0}$ sparse group lasso, where groups in the group lasso term could have overlapping group structure and tree structure [30]. From the theoretical point of view, we prove our algorithm always converges to the critical point (local optimal point) under mild conditions. In addition, we conduct extensive experiments on multiple public datasets and find MobilePrune achieves superior performance at sparsifying networks with both fully connected and convolutional layers. More importantly, we deploy our system on the real android system on several mobile devices and test the performance of the algorithm on multiple Human Activity Recognition (HAR) tasks. The results show that MobilePrune achieves much lower energy consumption and higher pruning rate while still retaining high prediction accuracy. Besides a powerful network compression algorithm, this work also provides a valuable platform and mobile dataset for further work to evaluate their methods in a very real scenario.

The rest of this paper is organized as follows. Section 2 provides the relevant background and related work. In Section 3, we give a brief overview of the proposed MobilePrune methods. In Section 4, we discuss the detailed information of the proposed methods and algorithms. In Section 5, we describe how the experiments are set up and evaluate the proposed methods from different perspectives. Section 6 discusses the future work and summarizes the paper.

## 2. Related Work

### 2.1. Sparsity for Deep Learning Models

Many pruning algorithms for deep learning models achieve slim neural networks by introducing sparse-inducing norms to the models. $\ell_{1}$ regularization [31,32,33] and $\ell_{0}$ regularization [8] were applied to induce sparsity on each individual weight. However, such individual sparsity has arbitrary structures, which cannot be utilized by software and hardware. Wei et al. [24] applied group sparsity to prune filters or channels, which can reduce the matrix size used in GEMM in cuDNN. Because the pruned models are compatible with cuDNN, they achieved large speedups. There are methods [34,35] aiming to find sparse models at both individual and group levels, which is similar to our goal. However, they all used $\ell_{1}$ norm to induce individual sparsity in addition to group sparsity. We have performed a comprehensive comparison and demonstrated that our MobilePrune method is the best in inducing sparsity at both individual and group levels for pruning deep learning models in Section 5.

### 2.2. Learning Algorithms for $\ell_{0}$ Norm

Recent studies [12,13] demonstrate that $\ell_{0}$ norm is the best sparsity-inducing penalty comparing to lasso $\ell_{1}$ [14], elastic net [15], SCAD [16], and MCP [17] models. Remarkably, even current $\ell_{0}$ optimization techniques cannot achieve the global optimal solution, these $\ell_{0}$-based methods with sub-optimal solutions still significantly outperform $\ell_{1}$ and elastic net models, which can be solved to global optima [13,18]. Therefore, the machine learning community has shown great interest in using $\ell_{0}$ norm to compress the large-scale DNN models [8,19,20,21,22,23,24]. However, some $\ell_{0}$-based DNN pruning approaches [8,25] rely on different relaxation strategies to conquer the non-convex and non-differentiable challenges, which does not fully exploit the strength of $\ell_{0}$ regularization. The other methods [19,20,21,22,23] rely on the alternating direction method of multipliers (ADMM) whose global convergence has not proved so far when applied to $\ell_{0}$ norm optimization [27,28,29].

### 2.3. Software & Hardware Compatibility

In this paper, we aim to design an algorithm to make the pruned DNN models compatible with cuDNN library [5] and hardware accelerator architecture that uses the sparse systolic tensor array [11,36]. cuDNN is the GPU-accelerated library used in deep learning models [5]. As shown in Figure 1a, convolution used in the convolutional neural network is lowered to matrix multiplication. Therefore, the size of the filter matrix can be reduced when inducing group sparsity column-wise as shown in Figure 1(b.3),(b.4), which can reduce the memory of the DNN models to achieve practical performance improvement. The systolic tensor array is an efficient hardware accelerator for structured sparse matrix as shown in Figure 1(c.4). Specifically, each column in Figure 1(c.4) is sparse. To achieve a pruned DNN model that is compatible with cuDNN and the systolic tensor array, sparsity needs to be induced on both the group level and within-group level. We will show how we achieve this in the following sections.

## 3. Overview

The central idea of MobilePrune is to compress the deep learning model in a way that is compatible with the architecture of data organization in the memory by combining $\ell_{0}$ regularization and group lasso regularization. The group lasso regularization helps to keep important groups of weights that benefit cuDNN, while $\ell_{0}$ regularization helps to achieve additional sparsity within those important groups that are needed for hardware acceleration. Figure 2 provides an overview of the proposed MobilePrune method. As illustrated in Figure 2a–c, the group lasso penalty will remove all the weights together with the *i*th neuron if it is less important for the prediction and if a group is selected, the $\ell_{0}$ penalty further removes the weights with small magnitudes within the group. Note that zeroing out the weights connected to the *i*th neuron results in removing the *i*th neuron and all the associated weights entirely. We will discuss more detail information in next section.

## 4. Methods

Our main objective is to obtain a sparse deep neural network with a significantly less number of parameters at both individual and group levels by using the proposed novel combined regularizer: $\ell_{0}$ sparse group lasso.

### 4.1. $\ell_{0}$ Sparse Group Lasso

We aim to prune a generic (deep) neural network, which includes fully connected feed-forward networks (FCN) and convolutional neural networks (CNN). Assuming the generic neural network has *N* neurons in FCN and *M* channels in CNN. Let $W^{i}$ denote the outgoing weights of the *i*th neuron in FCN and $T^{j}$ represent the 3D tensor of all filters in the *j*th channel, which can come from different layers. The training objective for the neural network is given as follows:(1)$$\begin{array}{cc} \min_{W,T}:L\left(W,T;D\right) & +\Omega_{\lambda }^{\eta }\left(W\right)+\Gamma_{\beta ,\gamma }^{\alpha }\left(T\right) \end{array}$$
where $W=\{W^{1},\ldots ,W^{N}\}$, $T=\{T^{1},\ldots ,T^{M}\}$, $D={\left\{x_{i},y_{i}\right\}}_{i=1}^{P}$ is a training dataset with *P* instances, L is an arbitrary loss function parameteized by *W* and *T*, $\Omega_{\lambda }^{\eta }\left(W\right)$ and $\Gamma_{\beta ,\gamma }^{\alpha }\left(T\right)$ represent the $\ell_{0}$ sparse group lasso penalties for neurons and channels, respectively. Specifically, $\Omega_{\lambda }^{\eta }\left(W\right)$ is defined as
(2)$$\begin{array}{c} \Omega_{\lambda }^{\eta }\left(W\right)=\sum_{i=1}^{N}(\eta ∥W^{i}{∥}_{0}+\lambda ∥W^{i}{∥}_{g}) \end{array}$$
where η≥0 and λ≥0 are regularization parameters. Let n(i) represent the set of outgoing edge weights of neuron *i*. Then, $∥W^{i}{∥}_{0}=\sum_{j\in n(i)}{∥W_{j}^{i}∥}_{0}$ ($W_{j}^{i}$ is the *j*th outgoing edge weight of neuron *i*; $∥W_{j}^{i}{∥}_{0}=0$ when $W_{j}^{i}=0$ and $∥W_{j}^{i}{∥}_{0}=1$ otherwise) computes the number of the non-zero edges in $W^{i}$ and $∥W^{i}{∥}_{g}=\sqrt{\sum_{j}{\left(W_{j\in n(i)}^{i}\right)}^{2}}$ aggregates the weights associated with the *i*th neuron as a group. The core spirit of Equation (Equation 2) is illustrated in Figure 2a–c, the group lasso penalty $∥W^{i}{∥}_{g}$ tends to remove all the weights together with the *i*th neuron if it is less important. If a group is selected, the $\ell_{0}$ penalty further removes the weights with small magnitudes within the group. Group sparsity $∥W^{i}{∥}_{g}$ can help remove neurons in the neural network, which reduces the size of the neural network and further improve efficiency. Individual sparsity $∥W^{i}{∥}_{0}$ helps to achieve additional sparsity within the remaining neurons. Such structured sparsity can be used by the systolic tensor array [10,11]. The other regularization term $\Gamma_{\beta ,\gamma }^{\alpha }\left(T\right)$ is defined as following,
(3)$$\begin{array}{c} \Gamma_{\beta ,\gamma }^{\alpha }\left(T\right)=\sum_{j=1}^{M}(\alpha ∥T^{j}{∥}_{0}+\beta ∥T^{j}{∥}_{g}+\gamma \sum_{h,w}∥T_{:,h,w}^{j}{∥}_{g}), \end{array}$$
where α,β, and γ are non-negative regularization parameters. Equation (Equation 3) defines a hierarchical-structured sparse penalty in which structure is guided by the memory organization of GEMM using cuDNN [5]. As demonstrated in Figure 2d, the pruning strategy encoded in Equation (Equation 3) explicitly takes advantage of the GEMM used in cuDNN. $∥T^{j}{∥}_{g}$ enforces the group sparsity of all the filters applied to the *j*th channel and $∥T_{:,h,w}^{j}{∥}_{g}$ enforces the group sparsity across the filters on the same channel and $∥T^{j}{∥}_{0}=\sum_{f}\sum_{h}\sum_{w}{∥T_{f,h,w}∥}_{0}$ prunes the small weights within the remaining channels and filters. Equation (Equation 3) can help to achieve an extremely compact model as Figure 1(c.4). Therefore, the computation can be accelerated at both software and hardware levels.

### 4.2. Exact Optimization by PALM

In this subsection, we first briefly review the general PALM (Proximal Alternating Linearized Minimization) framework used in our MobilePrune algorithm. Then, we introduce how we modify PALM to efficiently optimize the $\ell_{0}$ sparse group lasso for neural network compression. PALM is designed to optimize a general optimization problem formulated as:(4)$$\begin{array}{c} \min_{W,T}:F\left(W,T\right)+\Phi_{1}\left(W\right)+\Phi_{2}\left(T\right). \end{array}$$
where F(W,T) is a smooth function and $\Phi_{1}\left(W\right)$ and $\Phi_{2}\left(T\right)$ do not need to be convex or smooth, but are required to be lower semi-continuous. The PALM algorithm applies proximal forward–backward algorithm [37] to optimize Equation (Equation 4) with respect to *W* and *T* in an alternative manner. Specifically, at iteration k+1, the temporal values of $W^{\left(k+1\right)}$ and $T^{\left(k+1\right)}$ for the proximal forward–backward mapping are derived by solving the following sub-problems,
(5)$$\begin{array}{cc} W^{\left(k+1\right)} & \in \min_{W}:\left\{\frac{c^{k}}{2}{∥W-U_{i}^{\left(k\right)}∥}_{F}^{2}+\Phi_{1}\left(W\right)\right\}, \end{array}$$
(6)$$\begin{array}{cc} T^{\left(k+1\right)} & \in \min_{T}:\left\{\frac{d^{k}}{2}{∥T-V_{i}^{\left(k\right)}∥}_{F}^{2}+\Phi_{2}\left(T\right)\right\}, \end{array}$$
where $U_{i}^{\left(k+1\right)}=W_{i}^{\left(k\right)}-\frac{1}{c^{k}}\nabla_{W}F\left(W^{\left(k\right)},T^{\left(k\right)}\right)$ and $V_{i}^{\left(k+1\right)}=T_{i}^{\left(k\right)}-\frac{1}{d^{k}}\nabla_{T}F\left(W^{\left(k+1\right)},T^{\left(k\right)}\right)$. Additionally, $c^{k}$ and $d^{k}$ are positive constants. This optimization process has been proven to converge to a critical point when functions *F*, $\Phi_{1}$, and $\Phi_{2}$ are bounded [37]. We further extend the convergence proof in [37] and prove that the global convergence of PALM holds for training deep learning models under mild conditions. The detailed proof can be found in Appendix A.

To optimize Equation (Equation 1), we define two proximal operators for the two penalty terms as the following,
(7)$$\begin{array}{cc} \pi_{\lambda }^{\eta }\left(y\right) & \equiv \arg\min_{x}\left\{\frac{1}{2}{∥x-y∥}_{2}^{2}+\Omega_{\lambda }^{\eta }\left(x\right)\right\}, \end{array}$$
(8)$$\begin{array}{cc} \theta_{\beta ,\gamma }^{\alpha }\left(y\right) & \equiv \arg\min_{x}\left\{\frac{1}{2}{∥x-y∥}_{2}^{2}+\Gamma_{\beta ,\gamma }^{\alpha }\left(x\right)\right\}. \end{array}$$

Here, functions $\Omega_{\lambda }^{\eta }\left(\cdot \right)$ and $\Gamma_{\beta ,\lambda }^{\alpha }\left(\cdot \right)$ take vectors as inputs, which are equivalent to Equations (Equation 2) and (Equation 3) once we vectorize $W^{i}$ and $T^{j}$. The overall optimization process of MobilePrune is described in Algorithm 1. Once we can efficiently compute the optimal solution of $\pi_{\lambda }^{\eta }\left(\cdot \right)$ and $\theta_{\beta ,\lambda }^{\alpha }\left(\cdot \right)$, the computational burden mainly concentrates on the partial derivative calculation of functions H(·) and F(·), which is the same as training a normal DNN model.
**Algorithm 1** The framework of MobilePrune Algorithm.Initialize ${\left(W^{i}\right)}^{0},\forall i$, ${\left(T^{j}\right)}^{0},\forall j$ and $L_{r}$.**for**k=0,2,…**do**   **for** i=1,2,…,N **do**     $H_{k}^{i}=\nabla_{W_{k}^{i}}L\left(W_{k}^{i},W_{k}^{j\neq i},T_{k}\right)$.     $W_{k+1}^{i}\in \pi_{\lambda /L_{r}}^{\eta /L_{r}}\left(W_{k}^{i}-\frac{1}{L_{r}}H_{k}^{i}\right)$ by solving Equation (Equation 7).   **end for**   **for** j=1,2,…,M **do**     $F_{k}^{j}=\nabla_{T_{k}^{j}}L\left(W_{k+1},T_{k}^{j},T_{k}^{l\neq j}\right)$.     $T_{k+1}^{j}\in \theta_{\beta /L_{r},\gamma /L_{r}}^{\alpha /L_{r}}\left(T_{k}^{j}-\frac{1}{L_{r}}F_{k}^{j}\right)$ by solving Equation (8).   **end for****end for**

### 4.3. Efficient Computation of Proximal Operators

To the best of our knowledge, $\pi_{\lambda }^{\eta }\left(y\right)$ and $\theta_{\beta ,\gamma }^{\alpha }\left(y\right)$ defined in Equations (Equation 7) and (8) are novel proximal operators that have not been attempted before. Solving $\pi_{\lambda }^{\eta }\left(y\right)$ and $\theta_{\beta ,\gamma }^{\alpha }\left(y\right)$ is the key to apply MobilePrune in Algorithm 1. Therefore, this subsection elaborates the algorithmic contributions we made to efficiently calculate $\pi_{\lambda }^{\eta }\left(y\right)$ and $\theta_{\beta ,\gamma }^{\alpha }\left(y\right)$.

#### 4.3.1. Proximal Operator $\pi_{\lambda }^{\eta }\left(\cdot \right)$

The difficulty of solving $\pi_{\lambda }^{\eta }\left(y\right)$ in Equation (Equation 9) is that both ${∥x∥}_{g}$ and ${∥x∥}_{0}$ are not differentiable when the vector x=0. Furthermore, ${∥x∥}_{0}$ calculates the number of non-zeros in the vector $x\in R^{n}$ and there are $C\left(n,0\right)+C\left(n,1\right)+\ldots +C\left(n,n\right)=2^{n}$ (C(n,k) computes the number of non-zero patterns in *x* where *k* elements in *x* are not zeros) possible combinations, which indicates the brute-force method needs $2^{n}$ computations to find the global optimal solution. However, here we prove that the $\pi_{\lambda }^{\eta }\left(y\right)$ in Equation (Equation 9) can be efficiently solved by a O(nlog(n)) algorithm in a closed from. We illustrate the algorithm in Algorithm 2 and prove its correctness in Theorem 1. To the best of our knowledge, it is the first efficient algorithm that can calculate this novel proximal operator.

**Theorem** **1.**
*The proximal operator $\pi_{\lambda }^{\eta }\left(y\right)$ can be written as*

(9)
$$\begin{array}{c} \pi_{\lambda }^{\eta }\left(y\right)\equiv \arg\min_{x}\left\{f\left(x\right):=\frac{1}{2}{∥x-y∥}_{2}^{2}+{\lambda ∥x∥}_{g}+\eta {∥x∥}_{0}\right\} \end{array}$$


*The optimal solution of this proximal operator can be computed by Algorithm 2.*


**Algorithm 2** Efficient calculation of $\pi_{\lambda }^{\eta }\left(y\right)$
**Input:** A sorted vector *y*, such that $|y_{1}\left|\geq \right|y_{2}|\geq \ldots$
**Output: **

$x^{*}$


**for**

i=0,…,n

**do**
   **if** $∥\vec{y_{i}}{∥}_{g}\leq \lambda$ **then**     $U_{i}=\frac{1}{2}{∥y∥}_{2}^{2}$   **else**     $U_{i}=-\frac{1}{2}(∥\vec{y_{i}}{∥}_{g}{-\lambda )}^{2}+i\eta +\frac{1}{2}{∥y∥}_{2}^{2}$   **end if**
**end for**
Compute $k=\arg\min_{j}U_{j}$
**if**

$U_{k}\geq \frac{1}{2}{∥y∥}_{2}^{2}$

**then**
   $x^{*}=0$
**else**
   $x^{*}=(∥\vec{y_{k}}{∥}_{g}-\lambda )\frac{\vec{y_{k}}}{∥\vec{y_{k}}{∥}_{g}}$
**end if**



**Proof.** Without loss of generality, we assume $y={\left[y_{1},y_{2},\ldots ,y_{n}\right]}^{T}\in R^{n}$ is an ordered vector, where $|y_{1}\left|\geq \right|y_{2}\left|\geq \ldots \geq \right|y_{n}|$. Then, we define $\vec{y_{k}}=\left[y_{1},y_{2},\ldots ,y_{k},0,\ldots ,0\right]$, where the top *k* elements with largest absolute values are kept and all the rest elements are set to zeroes. We define another set $\Phi^{k}={\{x|∥x∥}_{0}=k,x\in R^{n}\}$ to represent all *n*-dimensional vectors with exact *k* non-zero elements. For any $x={\left[x_{1},x_{2},\ldots ,x_{n}\right]}^{T}\in \Phi^{k}$, we further define a mask function $e:e_{i}=1\left\{x_{i}=0\right\}$ to reveal the non-zero locations of *x*.Since we do not know how many non-zero elements remain in the optimal solution of Equation (Equation 9), we need to enumerate all possible *k* and solve n+1 sub-problems for all $x^{k}\in \Phi^{k}$ with k=0,…,n. For each *k*, the sub-problem is defined as
(10)$$\begin{array}{cc} \; & \min_{x^{k}\in \Phi^{k}}f\left(x^{k}\right):=\frac{1}{2}{∥x^{k}-y∥}_{2}^{2}+\Omega_{\lambda }^{\eta }\left(x^{k}\right)\\ \Leftrightarrow & \min_{x^{k}\in \Phi^{k}}\frac{1}{2}∥x^{k}-y^{k}{∥}_{2}^{2}+\frac{1}{2}{∥\bar{y^{k}}∥}_{2}^{2}+\Omega_{\lambda }^{\eta }\left(x^{k}\right) \end{array}$$Based on Lemma A4 in Section A.2, we observe that if $∥y^{k}{∥}_{g}\leq \lambda$, then $x^{k*}=0$ and $f\left(x^{k*}\right)=\frac{1}{2}{∥y∥}_{2}^{2}$. If $∥y^{k}{∥}_{g}>\lambda$, then $x^{k*}=(∥y^{k}{∥}_{g}-\lambda )\frac{y^{k}}{∥y^{k}{∥}_{g}}$ and the value of the objective function can be computed as
(11)$$\begin{array}{cc} f(x^{k*})= & \frac{1}{2}∥x^{k*}-y^{k}{∥}_{2}^{2}+\frac{1}{2}{∥\bar{y^{k}}∥}_{2}^{2}+\Omega_{\lambda }^{\eta }\left(x^{k*}\right)\\ = & -\frac{1}{2}(∥y^{k}{∥}_{g}{-\lambda )}^{2}+\eta k+\frac{1}{2}{∥y∥}_{2}^{2}. \end{array}$$Equation (Equation 11) tells us $f(x^{k*})$ is a function of $y^{k}$. The task of calculating the minimum value of function $f(x^{k*})$ is transformed into solving another optimization $-\frac{1}{2}(∥y^{k}{∥}_{g}{-\lambda )}^{2}+\eta k+\frac{1}{2}{∥y∥}_{2}^{2}$ with respect to $y^{k}$, which is equivalent to ask which of the *k* components of *y* can achieve the minimum value of *f*. Since we assume $∥y^{k}{∥}_{g}>\lambda$, the optimal solution is clearly to select the top *k* components with the largest value from *y*. Therefore we have $\vec{y_{k}}={arg min}_{y^{k}}-\frac{1}{2}(∥y^{k}{∥}_{g}{-\lambda )}^{2}+\eta k+\frac{1}{2}{∥y∥}_{2}^{2}$ and the corresponding $x^{k*}=(∥\vec{y_{k}}{∥}_{g}-\lambda )\frac{\vec{y_{k}}}{∥\vec{y_{k}}{∥}_{g}}$. Furthermore, the objective function value is $-\frac{1}{2}(∥\vec{y_{k}}{∥}_{g}{-\lambda )}^{2}+\eta k+\frac{1}{2}{∥y∥}_{2}^{2}$. Hence, problem (Equation 10) has a closed-form solution.    □

As shown in Algorithm 2, the heaviest computation is to sort the input vector *y*, therefore, the time complexity for solving Equation (Equation 9) is O(nlog(n)).

#### 4.3.2. Proximal Operator $\theta_{\beta ,\gamma }^{\alpha }\left(y\right)$

Similar as $\pi_{\lambda }^{\eta }\left(y\right)$, $\theta_{\beta ,\gamma }^{\alpha }\left(y\right)$ is the solution of the following optimization problem:(12)$$\begin{array}{cc} \min_{x}: & \kappa \left(x\right):=\frac{1}{2}{∥x-y∥}_{2}^{2}+{\alpha ∥x∥}_{0}+{\beta ∥x∥}_{g}+\gamma \sum_{i=1}^{d}{∥x_{G_{i}}∥}_{g}\\ s.t. & \;\;\overset{d}{\underset{i=1}{⋃}}G_{i}=\left\{1,2,\ldots ,n\right\},G_{i}\cap G_{j}=\emptyset ,\;\forall i,j \end{array}$$
where we assume $x,y\in R^{n}$. $G_{i}\subseteq \left\{1,\ldots ,n\right\},i$ is the index of a group and *d* represents the number of groups. Note that the grouping structures specified in Equation (Equation 12) is a special case of the grouped tree structures [30], where ${∥x∥}_{g}$ is the group lasso for the root of the tree and all the $∥x_{G_{i}}{∥}_{g}$ are the group lasso terms of its children. Notice that groups from the same depth on the tree do not overlap and furthermore ${∥x∥}_{0}=\sum_{i=1}^{d}{∥x_{G_{i}}∥}_{0}$. To simplify the notation, assuming ${∥x∥}_{0}\neq 0$ we define $h\left(x\right)=\frac{1}{2}{∥x-y∥}_{2}^{2}+\beta {∥x∥}_{g}$ that is a convex and differentiable and rewrite the problem as
(13)$$\begin{array}{cc} \min_{x}:\;\; & h\left(x\right)+\sum_{i=1}^{d}\Omega_{\gamma }^{\alpha }\left(x_{G_{i}}\right)\\ s.t.\;\; & x\neq 0,\overset{d}{\underset{i=1}{⋃}}G_{i}=\left\{1,2,\ldots ,n\right\},G_{i}\cap G_{j}=\emptyset ,\;\forall i,j\in \left\{1,\ldots ,d\right\}, \end{array}$$
where $\sum_{i=1}^{d}\Omega_{\gamma }^{\alpha }\left(x_{G_{i}}\right)=\sum_{i=1}^{d}\alpha ∥x_{G_{i}}{∥}_{0}+\gamma {∥x_{G_{i}}∥}_{g}$. We can use the proximal method [38] to find a solution $x^{†}$ of Equation (Equation 13). In the proximal method, we need to estimate the Lipschitz constant $L\left(x\right)=1+\frac{\beta }{{∥x∥}_{g}}$ and the partial derivative $\nabla_{x_{G_{i}}}h\left(x\right)=\left(1+\frac{\beta }{{∥x∥}_{g}}\right)x_{G_{i}}-y_{G_{i}}$. In addition, we need to use Algorithm 2 to solve $\pi_{\gamma }^{\alpha }\left(\cdot \right)$ for each group $x_{G_{i}}$. After obtaining $x^{†}$, we can find the solution of Equation (Equation 12) by comparing κ(0) with $\kappa (x^{†})$. If $\kappa \left(0\right)\leq \kappa \left(x^{†}\right)$, then the local optimal solution of Equation (Equation 12) is $x^{*}=0$, otherwise, $x^{*}=x^{†}$. We elaborate the algorithm for the proximal operator $\theta_{\beta ,\gamma }^{\alpha }\left(y\right)$ in the Algorithm 3. The major computation cost is the proximal method, therefore, the convergence rate of Algorithm 3 is O(1/k) [38].
**Algorithm 3** Efficient calculation of $\theta_{\beta ,\gamma }^{\alpha }\left(y\right)$**Input:**$L^{0}$ and $x^{0}$**Output:**$x^{*}$**for**l=0,1,2,…,k**do**   Let $L^{l}=1+\frac{\beta }{∥x^{l}{∥}_{g}}$ and $u=x_{G_{i}}^{l}-\frac{1}{L^{l}}\nabla_{x_{G_{i}}}h\left(x^{l}\right)$, then compute $x_{G_{i}}^{l}\in \pi_{\gamma /L^{l}}^{\alpha /L^{l}}\left(u\right),\;\forall i$ by applying Algorithm 2.**end for****if**$\kappa \left(0\right)\leq \kappa \left(x^{k}\right)$**then**   $x^{*}=0$**else**   $x^{*}=x^{k}$**end if**

## 5. Experimental Setup and Results

### 5.1. Performance on Image Benchmarks

In this subsection, we compared our proposed MobilePrune approach with other state-of-the-art pruning methods in terms of prune rate, computational costs, and test accuracy. We mainly compared our methods with structured pruning methods because DNN models pruned by non-structure pruning methods could not obtain practical speedup as shown in Figure 1. Notably, we only compared the results that can be reproduced by the source codes provided by the competing methods. First, we briefly summarized their methodology. PF [32] and NN slimming [33] were simple magnitude-based pruning methods based on l1 norm. BC [9], SBP [7], and VIBNet [39] cast the DNN pruning into probabilistic Bayesian models. C-OBD [40], C-OBS [2], Kron-OBD [40,41], Kron-OBS [2,41], and EigenDamage [42] are Hessian matrix-based methods. $\ell_{0}$ norm penalized method [8] and group lasso penalized method [24] are also well-known methods.

In our experiments, we use NVIDIA Corporation as the GPU and the number of cores of the CPU is 12. All the baseline models were trained from scratch via stochastic gradient decent(SGD) with a momentum of 0.9. We trained the networks for 150 epochs on MNIST and 400 epochs on CIFAR-10 and Tiny-ImageNet with an initial learning rate of 0.1 and weight decay of 5 × 10^−4^. The learning rate is decayed by a factor of 10 at 50, 100 on MNIST and at 100, 200 on CIFAR-10 and Tiny-ImageNet, respectively. The details of hyper-parameters for all experiments are summarized in Appendix B. We also provide the computational efficiency of our methods in Appendix C.

#### 5.1.1. MNIST Dataset

We first applied MobilePrune to prune the LeNet-300-100 and LeNet-5 [7,8,9] models on the MNIST dataset [43]. LeNet-300-100 is a fully-connected neural network model with three layers and 267 K parameters. LeNet-5 is comprised of two [20, 50] convolutional layers and two [800, 500] fully-connected layers with 431K parameters. Here, we compared with the state-of-the-art structured network compression algorithms [7,8,9] in terms of pruned accuracy, remaining parameters, pruned architecture, and FLOPs of the pruned models.

As shown in the top half of the MNIST dataset of Table 1, our model achieves the least number of neurons after pruning the LeNet-300-100 model and the lowest drop of the prediction accuracy 0.01% compared to other methods. More importantly, our pruned model achieves the lowest FLOPs. Note that the architecture of our pruned model is as compact as L0-sep [8], but is extremely sparse with only 5252 weights left. This additional sparsity would be critical when applying hardware acceleration [10,11] to our pruned model.

In addition, we compared with SSL on pruning the first two convolutional layers as done in [24] in Table A2. SSL has the same group lasso penalty term as ours but without $\ell_{0}$ norm regularization. As shown, our method decreases the sizes of the filters from 25 and 500 to 16 and 65, respectively, which dramatically lowers the FLOPs. In addition, the non-zero parameters in those remaining filters is very sparse in our model.

The bottom half of the MNIST dataset in Table 1 shows the performance comparison on pruning the LeNet-5 model. The LeNet-5 model pruned by our method achieves the lowest FLOPs (113.50 K) with the smallest predicting accuracy drop 0.01%. Moreover, our pruned model also has the smallest number of weights (around 2310). In addition, we compared with SSL on pruning the first two convolutional layers as done in [24]. SSL has the same group lasso penalty term as ours, but without $\ell_{0}$ norm regularization. More details about SSL can be found in Section C.2. As shown, our method decreases the sizes of the filters from 25 and 500 to 16 and 65, respectively, which dramatically lowers the FLOPs. In addition, the non-zero parameters in those remaining filters are very sparse in our model.

#### 5.1.2. CIFAR-10 Dataset

We further evaluated our method on pruning more sophisticated DNN architectures, VGG-like [44] and ResNet-32 [42,45] and widen the network by a factor of 4, on CIFAR-10 [46]. Similarly, we compared with the state-of-the-art structured pruning methods [2,7,32,39,40,41,42] in terms of various metrics. As shown in the middle of Table 1, the pruned VGG-like model obtained by our method achieves the lowest FLOPs with the smallest test accuracy drop. Similar to previous results, our pruned model is able to keep the smallest number of weights in comparison to other methods, the key for hardware acceleration [10,11]. As presented in Table 1, the pruned ResNet-32 model achieved by our method outperforms other pruned models in terms of pruned test accuracy and FLOPs. In addition, in terms of the remaining weights, our pruned model is at the same sparsity level as C-OBD [40] while our pruned accuracy outperforms C-OBD by a large margin.

#### 5.1.3. Tiny-ImageNet Dataset

Besides the experiments on MNIST and CIFAR-10 datasets, we further evaluated the performance of our method on a more complex dataset, Tiny-ImageNet [47], using VGG-19 [48]. Tiny-ImageNet is a subset of the full ImageNet, which consists of 100,000 images for validation. There are 200 classes in Tiny-ImageNet. We compared our method with some state-of-the-art methods [2,33,40,42] in Table 1. As shown in Table 1, the test accuracy of the pruned model derived from our method outperforms all the other methods by a significant margin, about 10%, except EigenDamage. Our proposed method obtains the same-level test accuracy as EigenDamage. However, our method achieves a much sparser DNN model with 1.16 million fewer weights than EigenDamage. Meanwhile, our pruned model achieves lower FLOPs.

### 5.2. Performance on Human Activity Recognition Benchmarks

To demonstrate the efficacy and effectiveness of our proposed MobilePrune method, we perform a series of compassion studies with other state-of-the-art pruning methods such as $\ell_{0}$ norm, $\ell_{1}$ norm, $\ell_{2}$ norm, group lasso, and $\ell_{1}$ sparse group lasso for all three datasets—WISDM [49,50], UCI-HAR [51,52], and PAMAP2 [53,54,55]. We evaluate the pruning accuracy and pruning rate of weights (parameters) and nodes for our proposed MobilePrune approach and all other state-of-the-art pruning methods using the same learning rate (0.0001) and the same number of epochs (150) for all three datasets. The pruning thresholds are 0.015, 0.005, 0.01 for the pruning methods in the WISDM, HCI-HAR, and PAMAP2 datasets, respectively. In addition, we evaluate the computational cost and battery consumption for our proposed method with all other state-of-the-art pruning methods as well. The details of the dataset descriptions and the hyper-parameters for all experiments are summarized in Appendix D.

#### 5.2.1. Performance on the Desktop

We use Google Colab [56] to build a PyTorch backend on the above datasets. The GPU for Google Colab is NVIDIA Tesla K80 and the number of cores of the CPU for Google Colab is 2. As shown in Table 2, if we only use $\ell_{0}$ norm penalty or $\ell_{2}$ norm penalty, there is no effect on neurons or channels pruning as expected for all three datasets. Similarly, if we only employ group lasso penalty, the pruned model still has more weights or nodes left. For the UCI-HAR dataset, $\ell_{1}$ norm penalty and $\ell_{1}$ sparse group lasso penalty cannot sparse down the model while for the other two datasets, these two penalties could achieve better sparsity, but cannot be better than MobilePrune approach. There exists a trade-off between the pruned accuracy and the pruning rate. As can be seen in Table 2, our MobilePrune method still has high pruned accuracy even if there are not too many parameters and nodes left. In addition, we compare our MobilePrune method with $\ell_{1}$ sparse group lasso penalty. The $\ell_{0}$ sparse group lasso model still significantly outperforms the $\ell_{1}$ sparse group lasso model in weights and nodes pruning, which demonstrates its superiority in pruning CNN models.

We also calculate the response delay and time saving percentage for all the above methods on the desktop platform. Response delay is the time needed for the desktop to run the pre-trained model after the raw input signal is ready. Here in Table 2, the response delay results are obtained after running 200 input samples. As can be seen in Table 2, MobilePrune could save up to 66.00%, 57.43%, 90.20% on response delay on WISDM, HCI-HAR, and PAMAP2 datasets, respectively.

Overall, if we apply MobilePrune method, the pruned CNN models can still achieve the best sparsity in terms of both neurons (or channel) and weights without loss of performance. Additionally, the results in Table 2 show that our MobilePrune method could achieve 28.03%, 46.83%, 3.72% on weight (parameter) sparsity, and 52.52%, 68.75%, and 10.74% on node sparsity for the WISDM, UCI-HAR, and PAMAP2 datasets, respectively.

#### 5.2.2. Performance of Mobile Phones

We evaluate the computational cost and battery consumption for our proposed MobilePrune approach with all other state-of-the-art pruning methods. In order to obtain the final results about how these models perform on today’s smartphone, we implement an Android Application using Android Studio on Huawei P20 and OnePlus 8 Pro. PyTorch Android API [57] is used here for running trained models on Android devices. Currently, the Android devices only support running machine learning models by using CPU only. For the Huawei P20, the CPU is Cortex-A73. For the OnePlus 8 Pro, it is using Octa-Core as its CPU. We also use the Batterystats tool and the Battery Historian script [58] to test the battery consumption.

Table 3 shows the response delay results and battery usage for our proposed method and all other state-to-the-arts pruning methods. Response delay is the time needed for the smartphone’s system to run the pre-trained model after the raw input signal is ready. Here, in Table 3, the response delay results are obtained after running 200 input samples and the battery consumption results are obtained after running 2000 input samples for each penalty in all three datasets. For the HCI-HAR dataset, our MobilePrune approach could save up to 40.14%, 22.22% on response delay and 34.52%, 19.44% on battery usage for Huawei P20 and OnePlus Pro 8, respectively, while the other pruning methods stay almost the same compared to the uncompressed version. For the WISDM and PAMAP2 datasets, $\ell_{0}$ norm penalty, $\ell_{2}$ norm penalty, and group lasso penalty cannot sparse down the model, and therefore they cannot provide any savings in both response delay and battery consumption. $\ell_{1}$ norm and $\ell_{1}$ sparse group lasso methods could provide better time saving and battery consumption saving compared to those three penalties, but they still cannot perform better than the MobilePrune method, which saves 61.94% and 88.15% in response time, and 37.50% and 36.71% in battery consumption for WISDM and PAMAP2 dataset, respectively, on Huawei P20. Additionally, it also saves 52.08% and 77.66% in response time, and 32.35% and 37.93% in battery consumption for WISDM and PAMAP2 dataset, respectively, on OnePlus 8 Pro. Overall, results in Table 3 demonstrate MobilePrune’s superiority in pruning HAR CNN models for battery usage and computational cost on today’s smartphone.

### 5.3. Ablation Studies

To demonstrate the efficacy and effectiveness of the $\ell_{0}$ sparse group lasso penalty, we performed a series of ablation studies on various DNN models. As shown in Table 4, if we only use $\ell_{0}$ norm penalty, there is no effect on a neuron or channel pruning as expected. Similarly, if we only employ the group lasso penalty, the pruned model still has more weights left. However, if we apply $\ell_{0}$ sparse group lasso, we can achieve pruned DNN models that are sparse in terms of both neurons (or channel) and weights. In addition, we compare our $\ell_{0}$ sparse group lasso model with $\ell_{1}$ sparse group lasso [59] on pruning DNN models. Table 4 shows their comparison on pruning various DNN models. More details can be found in Section A.3 and Appendix C. As shown in Table 4 and the results in supplementary, the $\ell_{0}$ sparse group lasso model significantly outperforms the $\ell_{1}$ sparse group lasso model in all aspects, which demonstrates its superiority in pruning DNN models.

## 6. Conclusions

In this work, we proposed a new DNN pruning method MobilePrune, which is able to generate compact DNN models that are compatible with both cuDNN and hardware acceleration. MobilePrune compress DNN models at both group and individual levels by using the novel $\ell_{0}$ sparse group lasso regularization. We further developed a global convergent optimization algorithm MobilePrune based on PALM to directly train the proposed compression models without any relaxation or approximation. Furthermore, we developed several efficient algorithms to solve the proximal operators associated with $\ell_{0}$ sparse group lasso with different grouping strategies, which is the key computation of our MobilePrune. We have performed empirical evaluations on several public benchmarks. Experimental results show that the proposed compression model outperforms existing state-of-the-art algorithms in terms of computational costs and prediction accuracy. MobilePrune has a great potential to design slim DNN models that can be deployed on dedicated hardware that uses a sparse systolic tensor array. More importantly, we deploy our system on the real android system on both Huawei P20 and OnePlus 8 Pro, and the performance of the algorithm on multiple Human Activity Recognition (HAR) tasks. The results show that MobilePrune achieves much lower energy consumption and higher pruning rate while still retaining high prediction accuracy.

There are other options to further compress the neural network models such as Neural Logic Circuits and Binary Neural Networks, which all use binary variables to represent inputs and hidden neurons. These two models are orthogonal to our methods, which means our pruning model could be adopted on Neural Logic Circuits, Binary Neural Networks and other neural network architectures designed for mobile systems. We will explore which mobile neural network could be better integrated with our network compression model in the future.

## Figures and Tables

**Figure 1 sensors-22-04081-f001:**
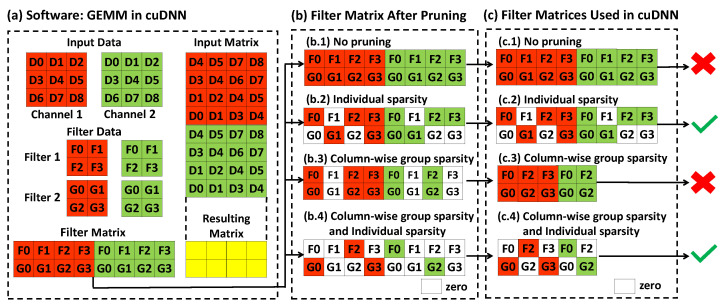
**Observations of different strategies’ pruned filter matrix for hardware acceleration with software implementation of convolution in cuDNN.** (**a**) General Matrix Multiply (GEMM) is applied in cuDNN. (**b**) Different strategies such as no pruning, individual sparsity, column-wise group sparsity, and both individual sparsity and column-wise group sparsity on pruning the filter matrix. (**c**) The pruned filter matrix implemented in cuDNN and determined whether it can be used for hardware acceleration or not.

**Figure 2 sensors-22-04081-f002:**
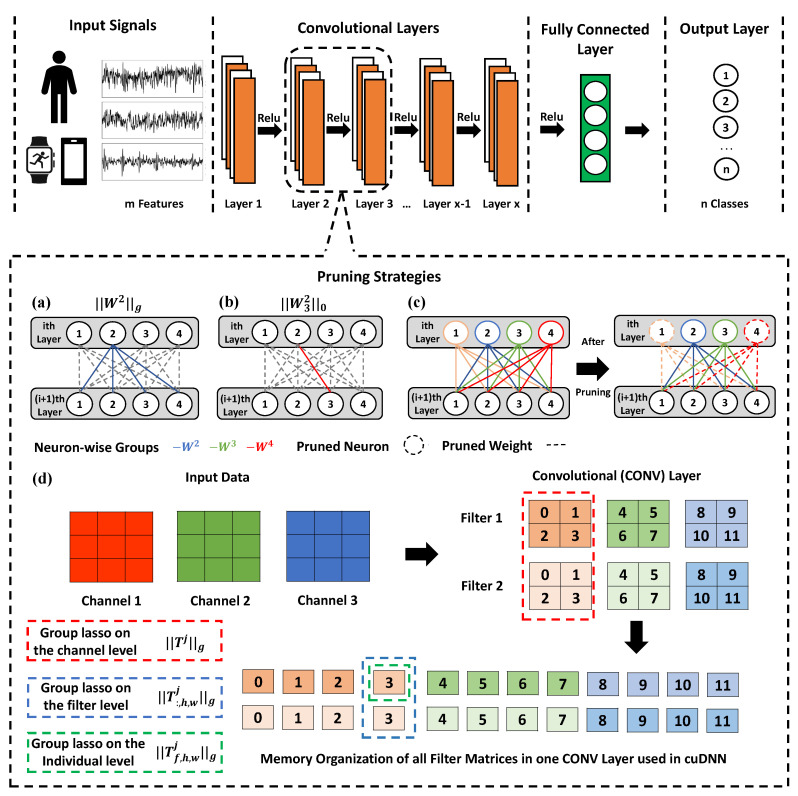
**Overview of the proposed MobilePrune method.** (**a**) Group sparsity for weights of a neuron for fully connected layers. (**b**) Sparsity on individual weights for fully connected layers. (**c**) Pruning strategy for fully connected layers and their effect where sparsity is induced on both neuron-wise groups and individual weights. (**d**) Group and individual sparsity for convolutional layers.

**Table 1 sensors-22-04081-t001:** Comparison of pruned models with state-of-the-art methods on different datasets – MNIST, CIFAR-10, and Tiny-ImageNet, respectively. (We highlight our MobilePrune results and mark the best performance as blue among different methods for each model in each dataset).

Dataset	Model	Methods	Base/Pruned Accuracy (%)	Original/Remaining Parameters (Mil)	FLPOs (Mil)
MNIST		BC-GNJ [9]	98.40/98.20	267.00/28.73	28.64
		BC-GHS [9]	98.40/98.20	267.00/28.17	28.09
	LeNet-300-100	L0 [8]	-/98.60	-	69.27
		L0-sep [8]	-/98.20	-	26.64
		**MobilePrune**	**98.24/98.23**	**267.00/5.25**	** 25.79 **
		SBP [7]	-/99.14	-	212.80
		BC-GNJ [9]	99.10/99.00	431.00/3.88	282.87
		BC-GHS [9]	99.10/99.00	431.00/2.59	153.38
	LeNet-5	L0 [8]	-/99.10	-	1113.40
		L0-sep [8]	-/99.00	-	390.68
		**MobilePrune**	**99.12/99.11**	**431.00/2.31**	** 113.50 **
CIFAR-10		Original [44]	-/92.45	15.00/-	313.5
		PF [32]	-/93.40	15.00/5.4	206.3
	VGG-like	SBP [7]	92.80/92.50	15.00/-	136.0
		SBPa [7]	92.80/91.00	15.00/-	99.20
		VIBNet [39]	-/93.50	15.00/0.87	86.82
		**MobilePrune**	**92.96/92.94**	**15.00/0.60**	** 77.83 **
		C-OBD [40]	95.30/95.27	7.42/2.92	488.85
		C-OBS [2]	95.30/95.30	7.42/3.04	378.22
	ResNet32	Kron-OBD [40,41]	95.30/95.30	7.42/3.26	526.17
		Kron-OBS [2,41]	95.30/95.46	7.42/3.23	524.52
		EigenDamage [42]	95.30/95.28	7.42/2.99	457.46
		**MobilePrune**	**95.29**/**95.47**	**7.42/2.93**	** 371.30 **
		NN slimming [33]	61.56/40.05	20.12/5.83	158.62
		C-OBD [40]	61.56/47.36	20.12/4.21	481.90
		C-OBS [2]	61.56/39.80	20.12/6.55	210.05
Tiny-ImageNet	VGG-19	Kron-OBD [40,41]	61.56/44.41	20.12/4.72	298.28
		Kron-OBS [2,41]	61.56/44.54	20.12/5.26	266.43
		EigenDamage [42]	61.56/56.92	20.12/5.21	408.17
		**MobilePrune**	**61.56**/**56.27**	**20.12/4.05**	**407.37**

**Table 2 sensors-22-04081-t002:** Comparison of pruning method on the desktop with state-of-the-art methods for pruning accuracy, pruning rate and response delay on HAR datasets—WISDM, HCI-HAR, and PAMAP2, respectively. (We highlight our MobilePrune results and mark the best performance as blue among different penalties for each dataset).

Dataset	Penalty	Base/Pruned Accuracy (%)	Parameter Nonzero (%)	Parameter Remaining (%)	Node Remaining (%)	Base/Pruned Response Delay (s)	Time Saving Percentage (%)
WISDM	$l_{0}$ norm	94.72/94.79	63.36	100.00	100.00	0.38/0.39	0.00
	$l_{1}$ norm	94.30/93.84	13.58	46.26	68.16	0.38/0.24	36.84
	$l_{2}$ norm	94.61/94.54	56.28	90.46	95.12	0.38/0.35	7.89
	Group lasso	94.68/94.32	48.23	89.73	94.73	0.38/0.35	7.89
	$l_{1}$ sparse Group lasso	94.81/94.79	17.91	53.41	73.83	0.41/0.26	36.59
	**MobilePrune**	**94.97/94.65**	** 9.52 **	** 28.03 **	** 52.52 **	**0.50/0.17**	** 66.00 **
UCI-HAR	$l_{0}$ norm	91.52/91.48	88.49	100.00	100.00	0.84/0.80	4.76
	$l_{1}$ norm	90.46/90.33	81.58	98.47	99.22	0.81/0.82	0.00
	$l_{2}$ norm	91.01/90.94	88.35	100.00	100.00	0.79/0.80	0.00
	Group lasso	90.80/90.84	82.91	100.00	100.00	0.83/0.78	6.02
	$l_{1}$ sparse Group lasso	91.11/91.04	81.21	97.70	98.83	0.84/0.80	4.76
	**MobilePrune**	**90.06/89.96**	** 23.00 **	** 46.83 **	** 68.75 **	**1.01/0.43**	** 57.43 **
PAMAP2	$l_{0}$ norm	93.15/93.07	69.27	100.00	100.00	0.41/0.41	0.00
	$l_{1}$ norm	95.22/95.29	1.46	7.28	19.73	0.40/0.08	80.00
	$l_{2}$ norm	92.08/92.09	65.32	94.93	97.27	0.41/0.39	4.88
	Group lasso	93.30/93.28	61.78	100.00	100.00	0.41/0.41	0.00
	$l_{1}$ sparse Group lasso	96.87/97.20	2.67	9.72	26.17	0.40/0.10	75.00
	**MobilePrune**	**96.89/96.95**	** 1.26 **	** 3.72 **	** 10.74 **	**0.51/0.05**	** 90.20 **

**Table 3 sensors-22-04081-t003:** Comparison of pruning method on the mobile devices with other state-of-the-art pruning methods for computational cost and battery usage on HAR dataset—WISDM, HCI-HAR, and PAMAP2, respectively. (We highlight our MobilePrune results and mark the best performance as blue among different penalties for each device in each dataset).

Dataset	Device	Penalty	Base/Pruned Response Delay (s)	Time Saving Percentage (%)	Based/Pruned Device Estimated Battery Use (%/h)	Battery Saving Percentage (%)
WISDM	Huawei P20	$l_{0}$ norm	1.40/1.27	9.29	0.71/0.70	1.41
		$l_{1}$ norm	1.33/0.71	46.62	0.74/0.65	12.16
		$l_{2}$ norm	1.28/1.21	5.47	0.74/0.77	0.00
		Group lasso	1.27/1.27	0.00	0.74/0.77	0.00
		$l_{1}$ sparse Group lasso	1.25/0.81	35.20	0.74/0.68	8.11
		**MobilePrune**	**1.34/0.51**	** 61.94 **	**0.72/0.45**	** 37.50 **
	OnePlus 8 Pro	$l_{0}$ norm	0.57/0.49	14.04	0.34/0.32	5.88
		$l_{1}$ norm	0.48/0.34	29.17	0.35/0.30	14.29
		$l_{2}$ norm	0.48/0.40	16.67	0.34/0.34	0.00
		Group lasso	0.49/0.45	8.16	0.34/0.35	0.00
		$l_{1}$ sparse Group lasso	0.48/0.33	31.25	0.35/0.30	14.29
		**MobilePrune**	**0.48/0.23**	** 52.08 **	**0.34/0.23**	** 32.35 **
HCI-HAR	Huawei P20	$l_{0}$ norm	1.43/1.43	0.00	0.84/0.84	0.00
		$l_{1}$ norm	1.42/1.42	0.00	0.85/0.84	1.18
		$l_{2}$ norm	1.43/1.43	0.00	0.84/0.84	0.00
		Group lasso	1.43/1.43	0.00	0.84/0.82	2.38
		$l_{1}$ sparse Group lasso	1.42/1.41	0.70	0.85/0.82	3.53
		**MobilePrune**	**1.42/0.85**	** 40.14 **	**0.84/0.55**	** 34.52 **
	OnePlus 8 Pro	$l_{0}$ norm	0.53/0.53	0.00	0.35/0.35	0.00
		$l_{1}$ norm	0.54/0.51	5.56	0.37/0.36	2.70
		$l_{2}$ norm	0.54/0.53	1.85	0.37/0.37	0.00
		Group lasso	0.53/0.52	1.89	0.36/0.36	0.00
		$l_{1}$ sparse Group lasso	0.53/0.52	1.89	0.36/0.36	0.00
		**MobilePrune**	**0.54/0.42**	** 22.22 **	**0.36/0.29**	** 19.44 **
PAMAP2	Huawei P20	$l_{0}$ norm	2.64/2.72	0.00	0.76/0.79	0.00
		$l_{1}$ norm	2.74/0.45	83.58	0.79/0.53	32.91
		$l_{2}$ norm	2.67/2.56	4.12	0.78/0.78	0.00
		Group lasso	2.67/2.68	0.00	0.78/0.78	0.00
		$l_{1}$ sparse Group lasso	2.69/0.55	79.55	0.79/0.57	27.85
		**MobilePrune**	**2.70/0.32**	** 88.15 **	**0.79/0.50**	** 36.71 **
	OnePlus 8 Pro	$l_{0}$ norm	0.94/0.93	1.06	0.88/0.88	0.00
		$l_{1}$ norm	0.93/0.25	73.12	0.87/0.55	36.78
		$l_{2}$ norm	0.93/0.91	2.15	0.88/0.87	1.14
		Group lasso	0.94/0.95	0.00	0.89/0.89	0.00
		$l_{1}$ sparse Group lasso	0.95/0.29	69.47	0.88/0.59	32.95
		**MobilePrune**	**0.94/0.21**	** 77.66 **	**0.87/0.54**	** 37.93 **

**Table 4 sensors-22-04081-t004:** Alation studies on various network models. (We mark the best performance as blue among different penalties for each model).

Network Model	Penalty	Base/Pruned Accuracy (%)	Original/Remaining Parameters (Mil)	FLOPs	Sparsity (%)
LetNet-300	$\ell_{0}$ norm	98.24/98.46	267 K/57.45 K	143.20	21.55
	Group lasso	98.24/98.17	267 K/32.06 K	39.70	12.01
	$\ell_{1}$ sparse group lasso	98.24/98.00	267 K/15.80 K	25.88	5.93
	$\ell_{0}$ sparse group lasso	98.24/98.23	267 K/5.25 K	25.79	1.97
LetNet-5	$\ell_{0}$ norm	99.12/99.20	431 K/321.0 K	2293.0	74.48
	Group lasso	99.12/99.11	431 K/8.81 K	187.00	2.04
	$\ell_{1}$ sparse group lasso	99.12/99.03	431 K/9.98 K	183.83	2.32
	$\ell_{0}$ sparse group lasso	99.12/99.11	431 K/2.31 K	113.50	0.54
VGG-like	$\ell_{0}$ norm	92.96/93.40	15 M/3.39 M	210.94	22.6
	Group lasso	92.96/92.47	15 M/0.84 M	78.07	5.60
	$\ell_{1}$ sparse group lasso	92.96/92.90	15 M/0.61 M	134.35	4.06
	$\ell_{0}$ sparse group lasso	92.96/92.94	15 M/0.60 M	77.83	4.00
ResNet-32	$\ell_{0}$ norm	95.29/95.68	7.42 M/6.74 M	993.11	90.84
	Group lasso	95.29/95.30	7.42 M/3.03 M	373.09	40.84
	$\ell_{1}$ sparse group lasso	95.29/95.04	7.42 M/5.66 M	735.12	76.28
	$\ell_{0}$ sparse group lasso	95.29/95.47	7.42 M/2.93 M	371.30	39.49
VGG-19	$\ell_{0}$ norm	61.56/61.99	138 M/19.29 M	1519.23	13.98
	Group lasso	61.56/53.25	138 M/5.93 M	683.99	4.30
	$\ell_{1}$ sparse group lasso	61.56/53.97	138 M/0.21 M	1282.82	0.15
	$\ell_{0}$ sparse group lasso	61.56/56.27	138 M/4.05 M	407.37	2.93

## Data Availability

We used the publicly available datasets—MNIST (http://yann.lecun.com/exdb/mnist/), CIFAR-10 (http://www.cs.toronto.edu/~kriz/cifar.html), Tiny-ImageNet (https://paperswithcode.com/dataset/tiny-imagenet), WISDM (https://archive.ics.uci.edu/ml/datasets/WISDM+Smartphone+and+Smartwatch+Activity+and+Biometrics+Dataset+), UCI-HAR (https://archive.ics.uci.edu/ml/datasets/human+activity+recognition+using+smartphones), and PAMAP2 (http://archive.ics.uci.edu/ml/datasets/pamap2+physical+activity+monitoring). Both the source codes for the proposed model results and mobile application can be found in https://github.com/yuboyubo/CNNCompressionl0 (accessed on 24 May 2022).

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
