# Peer review of "MobilePrune: Neural Network Compression via *ℓ*_0_ Sparse Group Lasso on the Mobile System"

_sensors, 2022, doi:10.3390/s22114081_

Round 1

Reviewer 1 Report

This paper introduces the MobilePrune algorithm capable of generating compact deep neural networks (DNN) models so that they run appropriately on mobile phones. Such algorithm is indeed original as it solves the proximal operator (pi^n_lamba (.)) for l_0 sparse group lasso in an exact manner. The frameworks defining the l_0 sparse group lasso, the PALM optimization, and the computation of the proximal operators are adequately explained. Experiments conducted with well-known datasets (MNIST, CIFAR-10, …) show outstanding results in prediction accuracy, computational cost, and battery consumption.

Overall, the paper represents an original contribution to the field of DNN compression devoted to mobile platforms. The paper is well written, it is easy to read, and to follow. I recommend its acceptance with minor remarks:

1. Line 152: it is not clear the meaning of: “Here we abuse the notations”

2. The experimental evaluation on mobile platforms is restricted to the Huawei P20. I wonder the performance on other platforms exhibiting GPUs of lower capacity. Some lines discussing computational cost according to the GPU would be appreciated.

3. I missed to read the future work perspectives of this research in the conclusion.

Reviewer 2 Report

MobilePrune is a good paper, and the compression rate method proposed in the paper has been effectively verified compared to other methods. The execution accuracy of the sparse group seems to be a little worse in some cases, but the generality of the reduced model in mobile can be applied to such cases.

Author Response

Response to Reviewer 2 Comments:

We thank the reviewer for the feedback. We agreed with the comments provided by the reviewer – “The execution accuracy of the sparse group seems to be a little worse in some cases, but the generality of the reduced model in mobile can be applied to such cases.” There always exists a tradeoff between accuracy and sparsity. Better sparsity means fewer nodes and weights and causes less accuracy sometimes. MobilePrune could handle such tradeoffs in great shape, and overall, the performance of MobilePrune is much better compared to the other state-of-the-arts.

Reviewer 3 Report

MobilePrune: Neural Network Compression via ℓ0 Sparse Group Lasso on the Mobile System

In this paper, the authors proposed a pruning method named MobilePrune that generates compact neural network models that can be deploy on desktop or resource-limited mobile platforms. This method is based on ℓ0 sparse group lasso regularization. The proposed method is claimed to outperform the existing state-of-the-art algorithms on the basis of computational costs and prediction accuracy. This is a hot research area and reduction in computational complexity of Deep Neural Networks can be beneficial for realization of DNN based real-time classification, recognition and segmentation models on embedded platforms.

The paper under discussion is well-written and easy to understand. Language of paper is technical and need no further changes. Introductory part is well defined and discusses different strategies to prune a filter matrix for accelerated matrix multiplication. Section 2 covers state of art related literature work and Section 3 and 4 presents proposed system in a proper way. However, following comments need to be addressed:

  • The specifications of both hardware should be mentioned and discussed in the results section (CPU Cores, GPUs, etc.) to better understand the results.

  • Table 2 presenting the Performance of proposed approach on the Desktop platforms should also list the response delay (inference time) and time saving percentage like comparison table of mobile phones as proposed approach is claimed to generate extremely compact neural network models for mobile platforms and as well as desktop platforms.

  • The proposed model should also be compared with CUDNN library in terms of inference time (response delay) for discussed datasets.

  • The implementation on Android application should also be discussed in detail as currently it is very brief. Integration of proposed approach in terms of used language (opencl, cuda) or library should also be mentioned.

Round 2

Reviewer 3 Report

Overall, all comments have been addressed. Thanks